

**Parametrizations of size distribution and refractive index of biomass burning organic**
**aerosol with black carbon content**
**Biao Luo[1,2], Ye Kuang[1,2,*], Shan Huang[1,2*], Qicong Song[1,2], Weiwei Hu[3], Wei Li[1,2], Yuwen Peng[1,2],**
**Duohong Chen[4], Dingli Yue[4], Bin Yuan[1,2], Min Shao[1,2]**
[1] Institute for Environmental and Climate Research, Jinan University, Guangzhou, China.
[2] Guangdong-Hongkong-Macau Joint Laboratory of Collaborative Innovation for Environmental
Quality, Guangzhou, China.
[3] State Key Laboratory of Organic Geochemistry and Guangdong Key Laboratory of Environmental
Protection and Resources Utilization, Guangzhou Institute of Geochemistry, Chinese Academy of
Sciences, Guangzhou 510640, China
[4] Guangdong Ecological and Environmental Monitoring Center, State Environmental Protection Key
Laboratory of Regional Air Quality Monitoring, Guangzhou 510308, China
Corresponding author: Ye Kuang (kuangye@jnu.edu.cn) and Shan Huang
(shanhuang_eci@jnu.edu.cn)



**Abstract**

Biomass burning organic aerosol (BBOA) impacts significantly on climate and regional air
quality directly through scattering and absorbing solar radiation and indirectly through acting as
cloud condensation nuclei. However, fundamental parameters in the simulation of BBOA radiative
effects and cloud activities such as size distribution and refractive index remain poorly
parameterized in models. In this study, biomass burning events were frequently observed during
autumn in the Pearl River Delta region, China. Aerosol physical properties including aerosol size
distributions, aerosol scattering coefficients and aerosol absorptions as well as aerosol chemical
compositions were comprehensively measured during these biomass burning events. An improved
absorption Ångström exponent (AAE) ratio method considering both variations and spectral
dependence of black carbon AAE was proposed to differentiate brown carbon (BrC) absorptions
from total aerosol absorptions. BBOA size distributions, mass scattering and absorption efficiency
were retrieved based on the changes in aerosol number size distributions, scattering coefficients and
derived BrC absorptions that occurred with BBOA spikes. Geometric mean diameter of BBOA
volume size distribution $D_{gv}$ depended largely on combustion conditions, ranging from 245 to 505
nm, and a linear relationship between $D_{gv}$ and $\Delta BC/\Delta BBOA$ was achieved. Retrieved BBOA mass
scattering efficiency, ranges from 3 to 7.5 $m^2/g$, depending nonlinearly on $D_{gv}$ (R=0.86) which was
confirmed by Mie theory simulations. Retrieved real part of BBOA refractive index ranges from
1.47 to 1.64, with evidences showing that its variations might depend largely on combustion
efficiency, which however requires further comprehensive investigations. Retrieved BBOA mass
absorption efficiencies and imaginary parts of BBOA refractive index ($m_{i,BBOA}$) correlated highly
with $\Delta BC/\Delta BBOA$ (R>0.88), but changes almost linearly with $\Delta BC/\Delta BBOA$ (R>0.88) which differs
much with previous findings. Consistent with results of previous studies, the variations of $m_{i,BBOA}$
as a function of optical wavelength $\lambda$ can be well parameterized using $m_{i,BBOA}(\lambda) =$
$m_{i,BBOA}(520) \times (\frac{\lambda}{520})^{w_{BBOA}}$. The spectral dependence parameter $w_{BBOA}$ ranged from 2.5 to 5.5 with
an average of 4.7 which is in generally higher than $w_{BBOA}$ values predicted by previous
parameterization schemes, however, is actually consistent with previous laboratory results of similar
$\Delta BC/\Delta BBOA$ ranges. In addition, $w_{BBOA}$ is also generally linearly correlated (R=-0.51) with
$\Delta BC/\Delta BBOA$. These findings have significant implications for simulating BBOA climate effects



and suggest that linking both BBOA refractive index and BBOA volume size dsitrbutions to black
carbon content might be a feasible and a good choice for climate models.






**1 Introduction**

Biomass burning organic aerosol (BBOA) emitted from natural and anthropogenic fire activities,

represents a major fraction of atmospheric primary organic aerosols, impacts significantly on climate
and regional air quality directly through scattering and absorbing solar radiation and indirectly through
acting as cloud condensation nuclei (Saleh et al., 2014;Saleh et al., 2015;Wang et al., 2016a;Zhang et
al., 2020;Liu et al., 2020b). BBOA size distributions are crucial for simulating aerosol-cloud
interactions, and BBOA scattering plays significant role in direct aerosol cooling effects and local
visibility degradation. BBOA is also a major contributor to atmospheric brown carbon (BrC) on a
global scale (Wang et al., 2016a) because of its non-negligible light absorption contribution in the near-
ultraviolet to visible wavelength. Accurate representation of BBOA size distributions, scattering and
absorption in climate models are crucial for BBOA radiative forcing simulations, and bias in biomass
burning absorption representation in models can result in biomass burning radiative forcing range from
cooling to warming (Brown et al., 2021). BBOA size distribution and refractive index are fundamental
parameters in the simulation of BBOA radiative effects and cloud activities, however, remain poorly
parameterized in models. Currently, our comprehensive knowledge of BBOA optical and physical
properties were primarily obtained from laboratory measurements (Janhäll et al., 2010;Saleh et al.,
2013;McClure et al., 2020). Although field measurements of biomass burning events were reported by
many studies (Laskin et al., 2015), however, only a few of them focused simultaneously on both BBOA
size distributions and optical properties (Reid et al., 2005b;Reid et al., 2005a;Laing et al., 2016), and
their parameterizations were reported by few studies. Comprehensive field measurements and
simultaneous characterization of BBOA size distributions, scattering and absorption properties and



retrieval of real and imaginary part of BBOA refractive index as well as their parameterizations remain
lacking, hindering the accurate representation of BBOA size distributions and refractive index in
climate models.
In-situ field measured aerosols are mixtures of different aerosol components emitted from
different sources and formed through different pathways. The BBOA mass concentrations might be
identified through source apportionment of organic aerosols using positive matrix factorization (PMF)
technique on the basis of aerosol mass spectrometer measurements (Kuang et al., 2021a). However,
the BBOA size distributions, BBOA scattering properties and BBOA light absorptions are usually quite
difficult to separate from properties of the entire aerosol populations. As a result, BBOA physical
properties such as size distribution, mass scattering efficiency (MSE), mass absorption efficiency
(MAE) and refractive index of biomass burning aerosols characterized in in-situ field measurements
are usually not specific to BBOA (Laing et al., 2016). Especially, parameterization of the imaginary
part of the BBOA refractive index ($m_{i,BBOA}$) have received wide attentions in recent years due to its
critical role in BBOA absorptivity representation in climate models (Saleh, 2020b). However, the yet
available parameterization schemes were primarily based on laboratory experiments, with very few
field measurements based results available (Lu et al., 2015). Liu et al. (2021) observed the evolution
of $m_{i,BBOA}$ in a real atmospheric environment chamber for different fire conditions at hourly scales
after emission under different oxidation conditions. Still, the spectral dependence parameterization of
$m_{i,BBOA}$ on the basis of in-situ field measurements covering a wavelength range from ultraviolet to
near-infrared remain lacking.
The key reason limiting the on-line characterization of BBOA refractive index based on the real
atmosphere measurements is that the on-line accurate quantification of BrC light absorption has been
a challenge due to the entanglement of black carbon (BC) absorption. Many studies have shown that
the distinct difference between BC and BrC spectral absorption characteristics represented by
Ångström law can be used to segregate BrC absorptions from measured total aerosol absorptions by
assuming a constant absorption Ångström exponent (AAE) of BC ($AAE_{BC}$) (de Sa et al., 2019;Wang
et al., 2016b;Yang et al., 2009). The BrC absorption retrieval accuracy of this constant AAE method
depends highly on the representativeness of used $AAE_{BC}$. Results of field and laboratory studies
demonstrated that $AAE_{BC}$ varies under different pollution and emission conditions (Zhang et al.,
2019a;Laskin et al., 2015). Model simulations and field observations show that $AAE_{BC}$ is affected by



many factors such as BC mixing state, morphology, BC mass size distribution as well as optical
wavelength, and values of $AAE_{BC}$ can reach up to 1.6 for specific wavelength pairs (Lack and Cappa,
2010). Recent studies have modified the AAE method through a better consideration of $AAE_{BC}$
variations. Zhang et al. (2019b) used the $AAE_{880-990}$ obtained from real-time aethalometer
measurements as $AAE_{BC}$, considering that aerosol absorptions at near infrared wavelengths are
associated only with BC. Other studies determined $AAE_{BC}$ through Mie theory simulations using
constrained BC mass or BC mixing states as inputs (Li et al., 2019;Wang et al., 2018;Qin et al.,
2018;Wang et al., 2016b). Wang et al. (2018) found remarkable $AAE_{BC}$ wavelength dependence and a
relatively stable ratio between $AAE_{BC}$ of certain wavelength ranges, which could be used to represent
spectral dependence of $AAE_{BC}$. However, this ratio method proposed by Wang et al. (2018) assumes
that BrC absorption contributes negligibly at 520 nm, which might bring some uncertainties and cannot
be used to retrieve the spectral characterization of BrC absorption for wavelengths near and beyond
520 nm.

In this study, aerosol chemical compositions, size distributions as well as aerosol scattering and

absorption coefficients were measured at a rural site in the Pearl River Delta (PRD) region of China,
where biomass burning events frequently occurred in autumn and played significant roles in regional
air quality (Liu et al., 2014). An improved method considering both variations and spectral dependence
of $AAE_{BC}$ was proposed to quantify the BrC absorption spectral dependence from 370 nm to 660 nm.
The differential method was applied to biomass burning events to estimate BBOA scattering and
absorption properties as well as BBOA size distributions. The combination of identified BBOA size
distributions, MSE and MAE were used to retrieve the real and imaginary parts of BBOA refractive
index using the Mie theory, based on which parameterizations of BBOA size distributions and
refractive index using BC/BBOA ratio were investigated.
**2 Materials and methods**
**2.1 Field measurements.**

Field measurements were performed from 30 September to 17 November 2019 at a rural site in

Heshan county, Guangdong Province, China. The site locates at the top of a small hill surrounded by
small villages and residential towns, and usually experiences air masses from cities of the highly
industrialized PRD region. This site is authorized as a supersite operated by the provincial





environmental monitoring authority, therefore continuous qualified measurements of meteorological parameters such as air temperature, relative humidity (RH), wind speed and direction, and pollutant measurements are carried out. Physical and chemical properties of ambient aerosol were comprehensively measured during this field campaign, including multi-wavelength aerosol scattering coefficients measurement under nearly dry (RH<30%) and controlled but fixed RH conditions using humified nephelometer system (Kuang et al., 2019;Kuang et al., 2021b), multi-wavelength absorption measurements using an aethalometer (Magee, AE33 (Drinovec et al., 2015)), aerosol size distribution measurements using a scanning mobility particle sizer (SMPS, TSI 3080) and an aerodynamic particle sizer (APS; TSI Inc., Model 3321), and aerosol chemical composition measurements using an soot-particle aerosol mass spectrometer, etc. The AE33 measurements were only valid from 30 September to 31 October. Continuous and stable measurements of aerosol chemical composition using the aerosol mass spectrometer measurements were valid since 10 October. More details on the site and instrument set up can be found in Kuang et al. (2021b).

Accurate AAE and absorption measurements are crucial for the BrC quantification. Results of previous comparison studies of aerosol absorption measurements between AE33 and photoacoustic soot spectrometer demonstrated that AAE will only be slightly influenced by the particle collection of AE33 on the filter (Saleh et al., 2013;Zhao et al., 2020). As to the absorption corrections associated with loading effect and multiple scattering effect caused by filter collection. Dual-spot mode was applied in AE33 measurements for dealing with aethalometer loading effect. A Multiple-scattering correction factor (C) was used to convert measured attenuation coefficient ($b_{ATN}$) by AE33 to the absorption coefficient of ambient aerosols ($b_{abs}$) at each wavelength through $b_{abs} = b_{ATN}/C$. C is considered to be dependent on filter tape, however, results of previous studied have reported that C might also varies with aerosol chemical compositions (Wu et al., 2009;Collaud Coen et al., 2010). The filter tape 8060 was used for AE33 during this field campaign. Zhao et al. (2020) evaluated C of filter tape 8060 through comparing AE33 measurements with a three-wavelength photoacoustic soot spectrometer, and their results demonstrated that C is almost independent of wavelength and differs little among measurements of different locations. Thus the wavelength independent C of filter tape 8060 of 2.9 recommended by Zhao et al. (2020) was used, and this value is also almost the median value of C ranges used in Kasthuriarachchi et al. (2020).



**2.2 Aerosol mass spectrometer measurements**.

The size-resolved aerosol chemical compositions of dried aerosol particles with aerodynamic

diameter less than 1 μm were measured using a soot particle aerosol mass spectrometer (SP-AMS,
Aerodyne Research, Inc., Billerica, MA, USA)(Kuang et al., 2021b). The mass concentrations of
aerosol chemical compositions from SP-AMS were validated by offline $PM_{2.5}$ filter measurements,
SMPS aerosol volume concentration measurements and online measurements for inorganic aerosol
components. More details on SP-AMS data quality assurance can be found in Kuang et al. (2021b).
The source identification of organic aerosols was conducted using positive matrix factorization (PMF)
method based on the high-resolution OA data collected in V-mode (only tungsten vaporizer). Six-
factors were identified based on the best performance criteria of PMF quality parameters. Two primary
OA factors include biomass burning organic aerosols (BBOA, O/C=0.48) and a hydrocarbon-like
organic aerosols (HOA, containing cooking emissions, O/C=0.02). The other four factors were
associated with secondary formations or aging processes: 1) more oxygenated organic aerosols
(MOOA, O/C=1, associated with regional airmass(Kuang et al., 2021b)), 2) less oxygenated organic
aerosols (LOOA, O/C=0.72, related to daytime photochemical formation), 3) nighttime-formed
organic aerosols (Night-OA, O/C=0.32, highly correlated with Nitrate with r=0.67, and exhibited sharp
increases during the evening), and 4) aged BBOA (aBBOA, O/C=0.39, exhibited similar diurnal
behavior with LOOA with strong daytime production). The mass spectral profile and time series of
these organic aerosol factors were shown in Fig.S3, and these factors were partly discussed in Kuang
et al. (2021b). The BBOA factor will be the focus of this study. On the basis of the scheme proposed
by Kuwata et al. (2012), the density of BBOA ($\rho_{BBOA}$) and HOA was estimated as 1.25 and 1.15 g/cm$^3$
with O:C and H:C as inputs, and used in this study.

**2.3 Quantification of BrC absorptions based on the light absorption wavelength dependence**
**measurements**.

BrC absorbs significantly at near-UV and short-visible wavelengths but exhibits strong

wavelength dependence (Saleh, 2020a). The deconvolution of the spectral dependence of measured
aerosol light absorption has been a common method to retrieve the BrC and black carbon (BC)
absorption distribution:
$\sigma_{BrC}(\lambda) = \sigma_a(\lambda) - \sigma_{BC}(\lambda)$ (1)



Where $\sigma_a(\lambda)$ represents measured total aerosol absorption at wavelength $\lambda$, $\sigma_{BC}(\lambda)$ the absorption
associated with BC (includes influences of BC size distributions and mixing states, etc.), and $\sigma_{Brc}(\lambda)$
the light absorption contributed by BrC. The spectral dependence of BC absorption was usually
accounted for using the Angstrom exponent (AAE) law (Laskin et al., 2015), which describes BC
absorption as $\sigma_{BC}(\lambda) = K\lambda^{-AAE}$ where K is a constant factor associated with BC mass concentration.
The traditional method usually assumes a constant AAE of 1(de Sa et al., 2019) , or a wavelength
independent AAE derived from near infrared absorption measurements by assuming that the BrC
absorption is negligible at near infrared wavelengths. For example,  $\sigma_{BC}(880\ nm)$ and $\sigma_{BC}(950\ nm)$
measured by AE33 can be used to formulate the spectral dependence of aerosol absorptions associated
with BC as the following:
$\sigma_{BC}(\lambda) = \sigma_{BC}(880\ nm) \times (\frac{880}{\lambda})^{AAE_{BC,\lambda-880}}$ (2)
$AAE_{BC,\lambda-880} = AAE_{BC,950-880}$ (3)

However, several recent modelling studies using Mie-theory and BC measurements demonstrated

that AAE$_{BC}$ varies as a function of wavelength, and the wavelength independent assumption of AAE$_{BC}$
will bring large uncertainties into BrC calculation (Li et al., 2019;Wang et al., 2018). Wang et al. (2018)
found  $AAE_{BC,520-880}$ and $AAE_{BC,370-520}$ differed much from each other, however, the
$AAE_{BC,370-520}/AAE_{BC,520-880}$ ratio varied little, and thus proposed an AAE ratio method to obtain
real-time $AAE_{BC,370-520}$ and further deduced $\sigma_{BC}(370\ nm)$ . This method assumes that BrC
contributes negligibly at 520 nm, which might introduce uncertainties. In addition, this method is not
applicable in retrieving the spectral dependence of BrC absorption because only the ratio
$AAE_{BC,370-520}/AAE_{BC,520-880}$ was used. This modified wavelength-dependent AAE differentiation
method was further partially adopted by Li et al. (2019),  using $AAE_{BC,370-520}$ to account for spectral
dependence of BC absorption for wavelengths<520 nm and $AAE_{BC,520-880}$ for wavelengths>520 nm,
thus the wavelength-dependent AAE$_{BC}$ was partially but not thoroughly considered.

Considering the advantages of both methods of  Wang et al. (2018) and Li et al. (2019), an

improved AAE ratio method was proposed to comprehensively tackle the spectral dependence of BC
absorption and also take real-time measured $AAE_{BC,950-880}$ into account, which combines the
modelled ratio R$_{AAE}$= $AAE_{BC,\lambda-880}$ / $AAE_{BC,950-880}$ and measured $AAE_{BC,950-880}$ to derive
$AAE_{BC,\lambda-880}$ and further retrieve $\sigma_{Brc}(\lambda)$ with the combination of Eq.1 and Eq.2. The modelling



method of $AAE_{BC,\lambda-880}$ is consistent with Li et al. (2019) and more details are available in Supplement
Sect. S1. The wavelength dependence of AAE$_{BC}$ are influenced by many factors such as BC refractive
index, coating shell refractive index as well as BC mixing state, and BC mass size distributions (Li et
al., 2019). A sensitivity experiment following the method of Li et al. (2019) is initiated to explore
impacts of these optical and mixing state parameters on $AAE_{BC,\lambda-880}$ and the ratio
$AAE_{BC,\lambda-880}/AAE_{BC,950-880}$. These parameters including the real part of the refractive index of BC
coating materials and BC-free particles (Real_NBC), real and imaginary parts of refractive index of
the BC core (Real_BC and Imag_BC), the mass fraction of externally mixed BC (r_ext), the number
fraction of BC-free particles (R_NBC), geometric standard deviation (GSD) and geometric mean
diameter (GM) of BC mass size distributions. Note that the imaginary parts of the refractive index of
BC particle coating materials and BC-free particles were not perturbed in these simulations and treated
as zero under the assumption of materials other than BC is non-absorbing. In order to separate effects
of BC and BrC on $AAE_{\lambda-880}$ changes, this assumption must be made to obtain $AAE_{BC,\lambda-880}$ variations
associated only with BC absorption changes. The defect of this method is that the entangling effects
of BrC coating on BC particles in $AAE_{BC,\lambda-880}$ variations are not considered. The results of
$AAE_{BC,370-880}$ is shown in Fig.1a. It shows that variations of both refractive index of BC and coating
materials as well as BC mixing states have non-negligible influences on $AAE_{BC,370-880}$, however the
BC mass size distributions represented by geometric standard deviation (GSD) and geometric mean
diameter (GM) of BC mass size distribution play the most important roles. Nevertheless, for results of
$AAE_{BC,\lambda-880}$ / $AAE_{BC,950-880}$ shown in Fig.1b, when fixing the BC mass size distribution,
$AAE_{BC,\lambda-880}/AAE_{BC,950-880}$ exhibited much smaller variations, even the refractive index of BC and
shell or mixing state varied within atmospherically relevant ranges. The result of sensitive studies
shown in Fig.1b further confirmed the applicability of the proposed new AAE ratio method under
constrained BC mass size distributions. The elemental carbon fragments (Cx) retrieved from SP-AMS
measurements cannot be used to quantify BC mass concentrations due to the lack of calibration
parameters, however, its size distributions generally represent the relative contributions of BC mass
within different diameter ranges. The real-time measured normalized Cx distributions are therefore
used to distribute total BC mass to different diameter bins to calculate the ratio





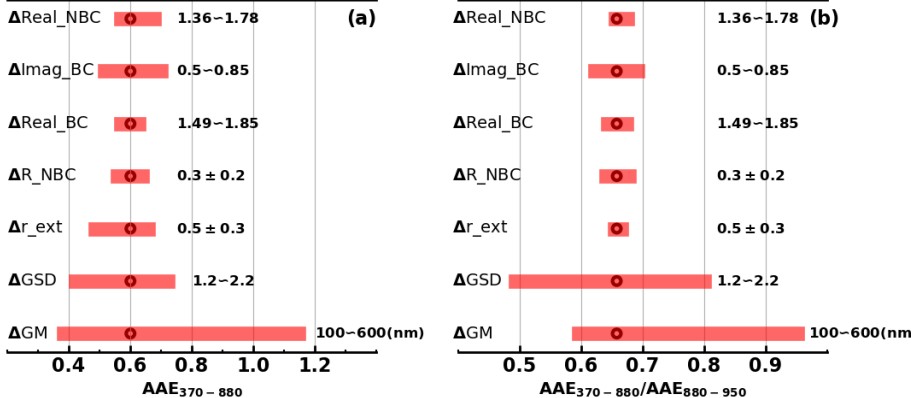

**Figure 1**. Changes in **(a)** $AAE_{BC,370-880}$ and **(b)** $AAE_{BC,370-880}/AAE_{BC,950-880}$ associated perturbations of different parameters.

$AAE_{BC,\lambda-880}/AAE_{BC,950-880}$, and the average normalized Cx distribution is shown in Fig.S4. The
average AAE ratios of $AAE_{BC,370-880}$ / $AAE_{BC,950-880}$, $AAE_{BC,470-880}$ / $AAE_{BC,520-880}$,
$AAE_{BC,590-880}$ / $AAE_{BC,950-880}$, $AAE_{BC,660-880}$ / $AAE_{BC,950-880}$, $AAE_{BC,370-880}$ / $AAE_{BC,950-880}$ are
0.79, 0.85, 0.88, 0.9 and 0.93 respectively. Based on this method, the spectral dependence of BrC
absorption can be derived as the following:
$\sigma_{BrC}(\lambda) = \sigma_a(\lambda) - \sigma_{BC}(880\ nm) \times (\frac{880}{\lambda})^{AAE_{BC,950-880} \times R(\lambda)}$ (4)
With this method, the effects of BrC coating on BC can still not be avoided, but the consideration of
aerosol absorptions associated only with BC would be improved than before.
**3 Results and discussions**
**3.1 Dominant contribution of BBOA to BrC absorption**
Biomass burning plumes around the observation site were frequently observed during this field
campaign at dusk as shown in Fig.S5(a,d) and only sometimes during daytime periods (Fig.S5(b, c)).
The average diurnal variations of resolved primary OA factors including both BBOA and HOA are
presented in Fig.S6, in which both average diurnal profiles of BBOA and HOA exhibited sharp
increases around 18:00 local time (LT), which should be associated with frequently observed biomass
burning events and supper cooking in villages and towns near this site. However, diurnal behaviors of
BBOA and HOA differ much from about 06:00 LT to 16:00 LT. HOA exhibited continuous decreases
during this daytime period, which was associated with boundary layer processes and re-partitioning





due to increasing temperature. The BBOA showed almost continuous but slow increases since morning
to the afternoon, indicating strong daytime emissions of BBOA as shown in Fig.S5(b, c), although not
as prominent as the BBOA emission just before the fall of nighttime. The probability distribution of
the ratio BBOA/HOA is also shown in Fig.S6b, which shows that the ratio BBOA/HOA reached
beyond 2 in 57% conditions with an average of 3.3, which demonstrates that biomass burning was a
dominant primary aerosol emission source during this field campaign.
The observed Angstrom Exponents between different wavelengths and 880 nm of total aerosol
absorption are shown in Fig.2a, the average values of $AAE_{370-880}$, $AAE_{470-880}$, $AAE_{520-880}$, $AAE_{590-880}$,
$AAE_{660-880}$, $AAE_{950-880}$ are 1.17, 1.23, 1.18, 1.15, 1.08, 1.04. The scatter plots of $AAE_{370-880}$ and the
ratio BBOA/BC shown in Fig.2b shows that $AAE_{370-880}$ was highly correlated with BBOA/BC (r=0.8),
indicating strong influences of BBOA on aerosol absorption wavelength dependence. The BrC
absorption at multiple wavelengths are extracted using the improved AAE ratio method introduced in
Sect.2, and statistical ranges of BrC absorption as well as their contributions to total aerosol absorption
are shown in Fig.2d. Average values of derived $\sigma_{BrC}$ at 370 nm, 470 nm, 520 nm, 590 nm, 660 nm are
$19.1\ Mm^{-1}$, $11.5\ Mm^{-1}$, $6.4\ Mm^{-1}$, $3.45\ Mm^{-1}$, $1.13\ Mm^{-1}$ and their contributions to total aerosol
absorption are 23%, 18%, 12%, 8%, 3% respectively. Similar to some previous studies (Tao et al.,
2020;Qin et al., 2018), these results shows that the contributions of BrC to aerosol absorption at
wavelengths of less than 590 nm are not negligible. The derived timeseries of $\sigma_{BrC,370}$ are shown in
Fig.S7d, depicting BBOA varying quite consistently with $\sigma_{BrC,370}$ and with high correlations
(correlation coefficients between $\sigma_{BrC}$ at 370 nm, 470 nm, 520 nm, 590 nm, 660 nm and BBOA
reaching 0.9, 0.83, 0.8, 0.76, 0.69), suggesting that BBOA was the dominant contributor to BrC
absorption.

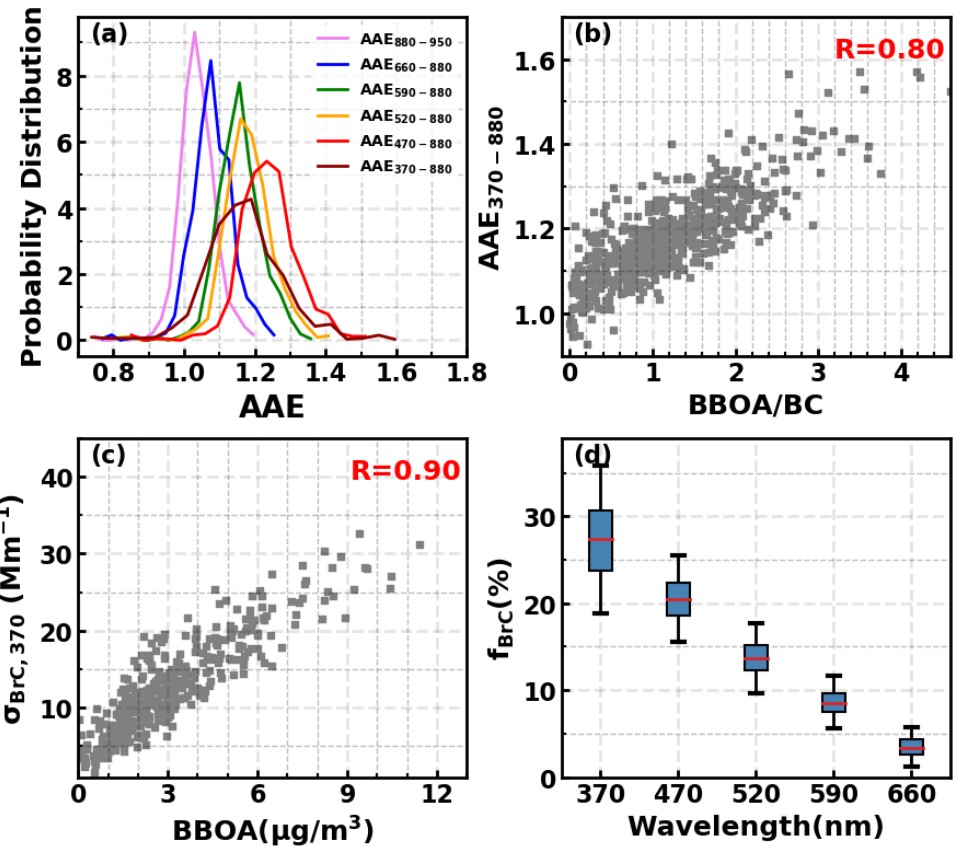

**Figure 2.** **(a)** Probability distribution of AAE between different wavelengths and 880 nm; **(b)** Correlations between $AAE_{370-880}$ and mass ratio of BBOA and BC; **(c)** Correlations between the BrC absorption coefficients at 370nm and the BBOA mass loadings; **(d)** Box-and-whisker plots of BrC absorption fractions at different wavelengths.


### 3.2 Identification of BBOA size distributions and their parameterizations

During the observation period, BBOA contributed domominantly to BrC absorptions and notable
biomass burning events represented by BBOA mass concentration spikes as shown in Fig.S7
freqeuntly occurred. These biomass burning spikes are related with biomass burning plumes that swept
over the observation site, thus the difference between aerosol properties measured before and during
these spikes can represent the properties of biomass burning aerosols. Theses spikes ususally occurred
during supper cooking time (~ 18:00 LT) and typical bio-fuels used for cooking are mainly vegetation



fuels such as local woods. SMPS directly measures the aerosol paritcle number size distribution
(PNSD), thus also providing particle volume size distribution measurements (PVSD). Fig.3a shows
the average differences of mass concentrations of different aerosol components of identified sipkes
with simultaneous valid SMPS data. Ammonium nitrate (AN) and ammonium sulfate (AS) were
determined as the dominant form of ammonium, sulfate and nitrate ions during this field campaign
and paired using the scheme paroposed by Gysel et al. (2007). It shows that inorganic aerosol
components increased a little bit, which is consistent with previous studies (Hecobian et al., 2011;Pratt
et al., 2011) that biomass burning emitts tiny amounts of inorganic aerosol. However, it is difficult to
quantity how much of these inorganic aerosol increases was attributed to biomass burning emissions
because the biomass burning spikes were usually observed during the periods with secondary nitrate
formation (Kuang et al., 2021a). Secondary organic aerosol components changed a little, with the slight
increase of aBBOA suggesting plumes were aged a little bit. Obvious increases of HOA were observed,
but the most prominent increase was BBOA. The aveage $\Delta BC/\Delta BBOA$ ratio for cases when BC
measurements were valid was 0.22, suggesting the observed biomass burning events are likely flaming
burning conditions with high combustion efficiency (Reid et al., 2005b;McClure et al., 2020). The
cooking related organic aerosol could not be separated from HOA in PMF analysis. The co-increase
of HOA are due to the fact that these identifed spikes occured during periods of supper cooking as
disccused before.

The average aerosol particle number and volume size distribution differences (Δ PNSD and Δ

PVSD) calculated as the PNSD and PVSD differnces between those at the BBOA peak concentration
and those before the BBOA spikes are shown in Fig.3b, the example of calculating Δ PNSD and Δ
PVSD is shown in Fig.S8. The average ΔPVSD can be well fitted using two lognormal modes (Mode
1 and Mode 2), the dominant one is BBOA and another is mostly associated with HOA according to
the aerosol mass changes. Geometric mean ($D_{gv}$) and standard deviation ($\sigma_g$) values of the two PVSD
lorgnormal modes are 180, 390  and 1.46, 1.5, respectively. In addition, the SP-AMS measurements



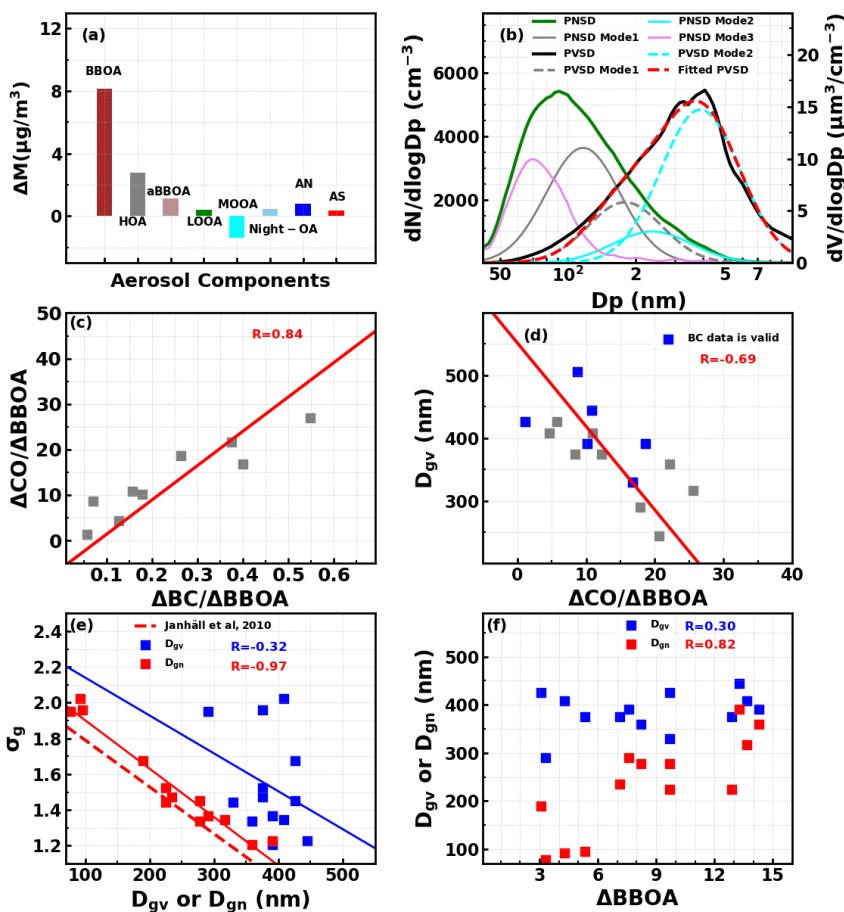

**Figure 3.** (a) Average differences of aerosol components before and end of BBOA spikes; (b) Corresponding particle average number and volume size distribution difference (ΔPNSD and ΔPVSD); (c) Relationship between ΔCO/ΔBBOA and ΔBC/ΔBBOA; (d) The relationships between identified $D_{gv}$ of BBOA spikes and corresponding ΔCO/ΔBBOA (ppb/(ug/m$^3$)); (e) relationship between retrieved $D_{gv}$ and $\sigma_g$, as well as $D_{gv}$ and $\sigma_g$. (f) relationships between $D_{gv}$ or $D_{gn}$ and ΔBBOA.

provides organic aerosol size distributions with vacuum aerodynamic diameter (Da), their average
distribution difference of organic aerosols during these spikes are also shown in Fig.S9 and could be
generally well fitted using two lognormal modes of BBOA and HOA. The $D_{gv,Da}$ and $\sigma_g$ values of the
identified modes were 175, 395 and 1.46, 1.55, respectively. $D_{va}$ and mobility diameter Dp of the
SMPS were related through the effective density of particles as $\rho_e = Da/(Dm \times C)$, where $\rho_e$ is the
aerosol effective density and C a factor related to aerosol shape, for which a value of 0.8 was adopted
(Jayne et al., 2000). Based on densities of BBOA and HOA introduced in Sect.2.2, identifed $D_{gv}$ of
BBOA and HOA from SP-AMS measurements of 395 and 190 nm, which were quite close to the $D_{gv}$



identified from SMPS measurements, further confirming the results from SMPS measurements. The
average ΔPNSD is shown in Fig.3b, displaying a number concentrations peak near 90 nm, however,
influences of HOA need to be excluded to identify biomass burning PNSD modes. As shown in Fig.3b,
converting the identified BBOA and HOA ΔPVSD modes to ΔPNSD modes cannot explain the
observed PNSD difference, the remaining mode is lorgnormal and peaks at 70 nm. These results
indicate that two modes existed for biomass burning aerosols during this campaign, which is consistent
with findings of previous studies (Okoshi et al., 2014;Liu et al., 2020a).

For spikes where ΔBBOA dominated the mass changes, the $D_{gv}$ and $\sigma_g$ of BBOA PVSD was

retrieved by fitting the larger mode of Δ PVSD, with retrieved results shown in Fig.3c and Fig.3d. The
retrieved $D_{gv}$ ranged from 245 nm to 505 nm with an average of 380 nm. Physicochemical properties
of biomass burning emissions depended largly on combustion conditions. BC/BBOA ratio is a proxy
of biomass combustion efficiencies (McClure et al., 2020), and it was found that ΔCO/ΔBBOA was
highly correlated with ΔBC/ΔBBOA (Fig.3c, R=0.84). Thus, ΔCO/ΔBBOA was also used as a proxy
for combustion efficiency in this study. Higher ΔCO/ΔBBOA corresponds to higher combustion
efficiency.  Retrieved $D_{gv}$ values were moderately but negatively correlated with ΔCO/ΔBBOA (R=-
0.69), and a linear relationship $D_{gv}$=551-13.3×ΔCO/ΔBBOA was derived. This result is qualitively
consistent with previous studies that biomass burning aerosols were mainly in the accumulation mode
and their average sizes generally decreased as the combustion efficiency increases (Reid and Hobbs,
1998;Janhäll et al., 2010). Retrieved $\sigma_g$ ranges from 1.2 to 2.0 with an average of 1.5, and is negatively
and weakly correlated with $D_{gv}$ (R=-0.32). Reid et al. (2005b) reported that $D_{gv}$ is typicaly in the range
of 250 to 300 nm with the $\sigma_g$ on the order of 1.6 to 1.9 for freshly generated smoke, and 30-80 nm
larger for aged smoke with smaller $\sigma_g$ (1.4 to 1.6). Levin et al. (2010) performed laboratory
combustion of various wildland fuels, and reported $D_{gv}$ of 200 to 570 nm and $\sigma_g$ of 1.68 to 2.97. The
average $D_{gv}$ and $\sigma_g$ is near the reported $D_{gv}$ range by Reid et al. (2005b) for aged smoke. Geometric
mean of PNSD ($D_{gn}$) values are converted from retrieved $D_{gv}$ and $\sigma_g$ and also shown in Fig.3e. $D_{gn}$
ranges from 88 to 391 nm with an average of 235 nm. The average $D_{gn}$ is similar with the reported
aveage $D_{gn}$ of aged smoke but the range even beyond the range (100-300 nm) for both fresh and aged
smokes reported by Janhäll et al. (2010) in which literature published $D_{gn}$ are reviewed, and also
beyond the range (about 130-240 nm) reported in Laing et al. (2016) for aged biomass burning aerosol





from wildfires in Siberia and the Western USA. Similar with resuls of Janhäll et al. (2010), $\sigma_g$ is highly
but negatively correlated with $D_{gn}$ (R=-0.97). The derived linear relationship $\sigma_g$=2.17-0.0027×$D_{gn}$ is
close to that reported in Janhäll et al. (2010) (Fig.3e). Janhäll et al. (2010) defined the fresh smoke as
plumes younger than 1 h, but aged smoke are mostly plumes older than one day. The aged smoke in
Laing et al. (2016) were also transported over 4-10 days. However, the smoke plumes reported in this
stduy occurred during supper cooking time, and swept over the observation site last about 1-3h (from
the begining to BBOA concentration fall back the background levels) which are consistent the time
need for cooking, which means that the age of plumes are on ther oder of hour and near freashly
emmited. This is indirectly confimed by the observed changes in particle number concentrations that
small aitken mode dominate the particle number contrations (Fig.3b), bacuase coagulation is quick and
should cause a significant decrease in number concentrations of Aitken mode aerosols in times scales
of hours (Sakamoto et al., 2015;Laing et al., 2016;Sakamoto et al., 2016). These results demonstrate
that $D_{gn}$ and $D_{gv}$ varies over a wide range for near freshly emitted BBOA from vegetation fire smokes.
Laing et al. (2016) reported that $D_{gn}$ was highly correlated with plume aerosol mass concentrations
(PM), but not with any normalized variable such as ΔPM/ΔCO. Simlar results were obtained in this
study (Fig.3f). The derived $D_{gn}$ was weakly correlated (R=-0.21) with ΔCO/ΔBBOA, but highly
correlated with ΔBBOA (R=0.82). The new finding here is that $D_{gv}$ correlated obviously with
ΔCO/ΔBBOA, but weakly with ΔBBOA. As discussed in implications, BBOA volume size
distributions determine BBOA bulk optical proeprties thus accurate representations of BBOA volume
size dsitrutbuions in climate models might be more important than accurate representations of BBOA
number size dsitrutbuions.

**3.3 BBOA Mass Scattering Efficiency and retrieval of the real part of BBOA refractive index**

The measured aerosol scattering coefficients at 525 nm ($\sigma_{sp,525}$) during BBOA spikes were used

to calculate the MSEs using the differential method, thereby retrieving the real part of BBOA refractive
index ($m_R$) on the basis of Mie theory. Truncation error, non-ideality of light source and RH conditions
need to be corrected in the calculation of $\sigma_{sp,525}$ values under dry condition. The truncation error and
non-ideality of light source was corrected using the empirical formula provided by Qiu et al. (2021).
$RH_0$ in the dry nephelometer was in the range of 20% to 45% with an average of 31%, and corrected



by considering measured aerosol optical hygroscopicity through $\sigma_{sp,525}=\sigma_{sp,525,measured}/(1+\kappa_{sca} \times$
$\frac{RH_0}{100-RH_0}$), where $\kappa_{sca}$ is the optical hygroscopicity parameter derived from aerosol light scattering
enhancement factor measurements (Kuang et al., 2017). To quantify $MSE_{BBOA}$, MSEs of other aerosol
components are needed. Using the paired campaign average size distributions of AS and AN (Fig.S1),
MSEs of AS and AN was calculated as 4.6 and 4.8 $m^2/g$, which were identical with those identified by
Tao et al. (2019) during autumn at an urban area in this region, but much higher than average values
reported in Hand and Malm (2007). Through the analysis of the OA distribution measured by SP-AMS,
it was found that the size distribution of SOA can be represented by two lognormal modes (Fig.S2).
One is aBBOA, and the other one includes MOOA, Night-OA, and MOOA. Thus, MSE of MOOA,
Night-OA, LOOA ($MSE_{SOA}$ ) was determined to be 6.3 $m^2/g$, and $MSE_{aBBOA}$ was 4.5 $m^2/g$. $MSE_{HOA}$
was calculated to be 3.2 $m^2/g$ using the size distribution identified in Fig.3b. $MSE_{BC}$ was calculated as
2.8 $m^2/g$ using the average normalized Cx fragments distributions, which was also very close to the
MSE of elemental carbon determined by Tao et al. (2019) (2.6 $m^2/g$). The changes of aerosol scattering
coefficients associated only with BBOA can be calculated as $\Delta\sigma_{sp,BBOA}=\Delta\sigma_{sp,measured}$ -$\Delta AS \times$
$MSE_{AS}$-$\Delta AN \times MSE_{AN}$-$\Delta HOA \times MSE_{HOA}$-$\Delta BC \times MSE_{BC}$-$\Delta aBBOA \times MSE_{aBBOA}$–($\Delta$Night-
OA+$\Delta$MOOA+$\Delta$LOOA)$\times MSE_{SOA}$. More details about MSE calculations of these components can be
found in Sect.S1. In addition, to minimize the influences of uncertainties of used MSEs of other aerosol
components on $MSE_{BBOA}$ derivations, only spikes with sum changes of $\Delta AS$, $\Delta AN$, $\Delta$Night-OA,
$\Delta$MOOA, $\Delta$LOOA and $\Delta$aBBOA accounting for less than 25% of $\Delta$ BBOA were used. Average changes
of aerosol components for these spikes are shown in Fig.4a, with changes of most individual aerosol
components being almost negeligible.

As shown in Fig.4b, the derived $\Delta\sigma_{sp,525}$ associated with BBOA was highly correalted with $\Delta$

BBOA (R=0.91). $MSE_{BBOA}$ ranged from 3.1 to 7.5 $m^2/g$ with an average of  5.3 $m^2/g$. Reid et al. (2005a)
reviewed the MSEs of biomass burning ($MSE_{BB}$) aerosols and reported a range of 3.2-4.2 $m^2/g$ for
temperate and boreal fresh smoke, and larger for corresponding aged smoke (4.3 $m^2/g$). McMeeking
et al. (2005) theoretically calculated the MSEs of smoke-influenced aerosols and reported a MSE range
of 3-6 $m^2/g$.  Levin et al. (2010) conducted $MSE_{BB}$ measurements of fresh biomass burning smokes of
various fuel types, reported a $MSE_{BB}$ range of 1.6 to 5.7 $m^2/g$. Laing et al. (2016) reported a $MSE_{BB}$
range of 2.5 to 4.7 for aged biomass burning aerosols of wildfires, and similar range was reported by



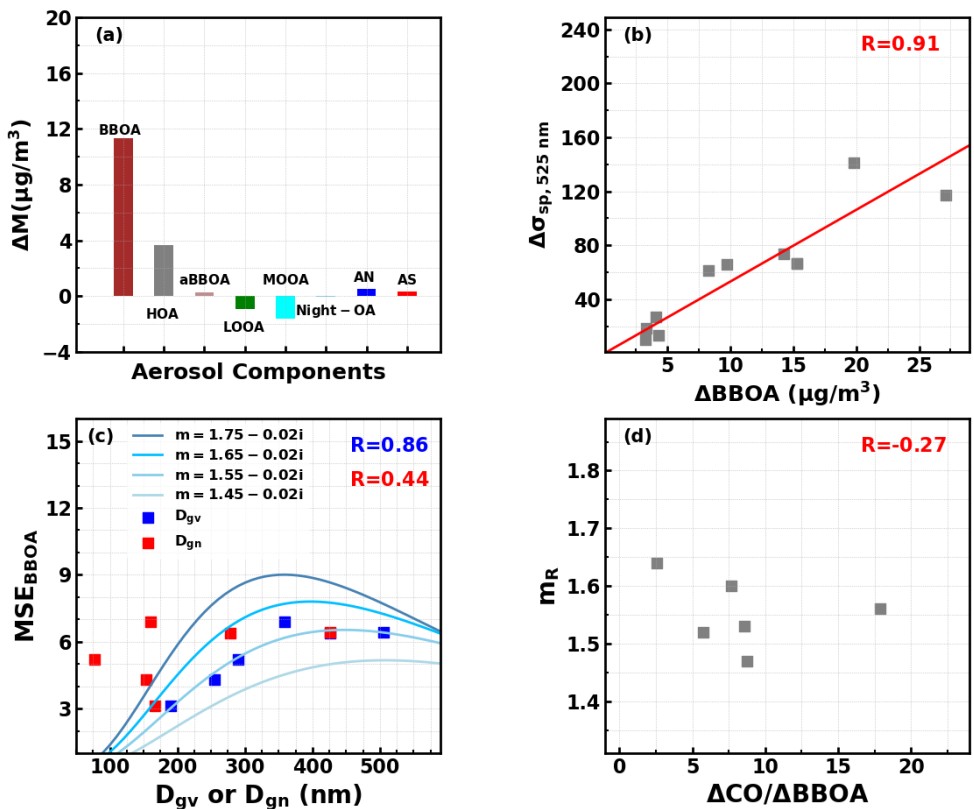

**Figure 4.** **(a)** Average differences of aerosol components between nearest background and the peak of BBOA spikes; **(b)** Relationships between derived $\Delta\sigma_{sp}$ at 525 nm only associated with BBOA and $\Delta$ BBOA; **(c)** Relationships between retrieved $MSE_{BBOA}$ and $D_{gn}$ or $D_{gv}$; **(d)** Relationship between retrieved $m_R$ and $\Delta CO/\Delta BBOA$.

Briggs et al. (2017). However, no study has specifically investigated $MSE_{BBOA}$ variations, which are
very crucial for biomass burning aerosol climate effects simulations, since aerosol components in
models are usually separately represented (Riemer et al., 2019). Although organic aerosols usually
dominate mass concentration of biomass burning aerosols, the reported $MSE_{BBOA}$ range is generally
higher than previously reported $MSE_{BB}$ ranges, which are likely associated with the fact that $MSE_{BB}$
includes influences of low scattering efficiency components such as BC. Another reason for this is that
the identified geometric mean size of BBOA in this study was generally larger than those reported
before. Many studies have shown that aerosol size distribution have crucial impacts on MSE
variations(Hand and Malm, 2007). Both results of Levin et al. (2010) and Laing et al. (2016) have
reported that $MSE_{BB}$ of biomass burning aerosols were highly correlated with $D_{gn}$. The relationship



between MSE$_{BBOA}$ and D$_{gn}$ as well as D$_{gv}$ were investigated (Fig.4c, only six points with both D$_{gv}$ and $\sigma_g$ retrieval are available). Unlike results of previous studies, MSE$_{BBOA}$ were positively but weakly correlated with D$_{gn}$ (R=0.44). However, MSE$_{BBOA}$ were highly correlated to D$_{gv}$(R=0.86), and exhibited non-linear response with the increase of D$_{gv}$. The non-linear increase phenomenon was reported first but confirmed by Mie theory simulations by assuming a fixed $\sigma_g$ of 1.5 under varying conditions of D$_{gv}$ and refractive index (Fig.4c).

Aerosol refractive index was a fundamental parameter in simulating aerosol optical properties in models. However, aerosol refractive index investigations specific to BBOA is scarce because the direct retrieval of aerosol refractive index at least needs accurate and simultaneous representations of MSE$_{BBOA}$, BBOA density and BBOA size distribution shape. Only few studies have indirectly retrieved m$_R$ of biomass burning related aerosols. For example, McMeeking et al. (2005) and Levin et al. (2010) have retrieved m$_R$ of biomass burning or smoke-influenced aerosols through using an iterative algorithm to match measured size distributions of different principles (mobility-related size versus optical size), reported m$_R$ ranges were 1.56 to 1.59 and 1.41 to 1.61, respectively. In this study, m$_R$ values of BBOA were retrieved using Mie theory with MSE$_{BBOA}$, D$_{gn}$, $\sigma_g$ and BBOA density as inputs as introduced in Sect.1.3 of the supplement. Note that the retrieval of m$_R$ would also be affected by the imaginary part of BBOA refractive index (m$_{i,BBOA}$), and the m$_{i,BBOA}$ parameterization as a function of $\Delta$CO/$\Delta$BBOA introduced in the next section was used. Retrieved m$_R$ ranges from 1.47 to 1.64 with an average of 1.56. If m$_R$ changes from 1.47 to 1.64 can result in a double MSE$_{BBOA}$ for given BBOA size distributions. Thus, reported m$_{R,BBOA}$ range was wide with respect to MSE simulations and needs to be carefully parameterized in climate modes. BBOA refrative index is determined by its chemical structure thus its variation might be associated with fire combustion conditions. The relationship between m$_{R,BBOA}$ and $\Delta$CO/$\Delta$BBOA was further investigated and shown in Fig.4d. For $\Delta$CO/$\Delta$BBOA below 10 ppb/$\mu g \cdot m^3$, m$_R$ was negatively correlated with $\Delta$CO/$\Delta$BBOA (R=-0.71) thus like $\Delta$BC/$\Delta$BBOA, which however, was not as significant (R=-0.27). These results demonstrate that fire combustion conditions might have significant impacts on m$_{R,BBOA}$, however, needs further investigation.

**3.4 BBOA mass absorption efficiency and parameterizations of the spectral dependence of imaginary part of BBOA refractive index**



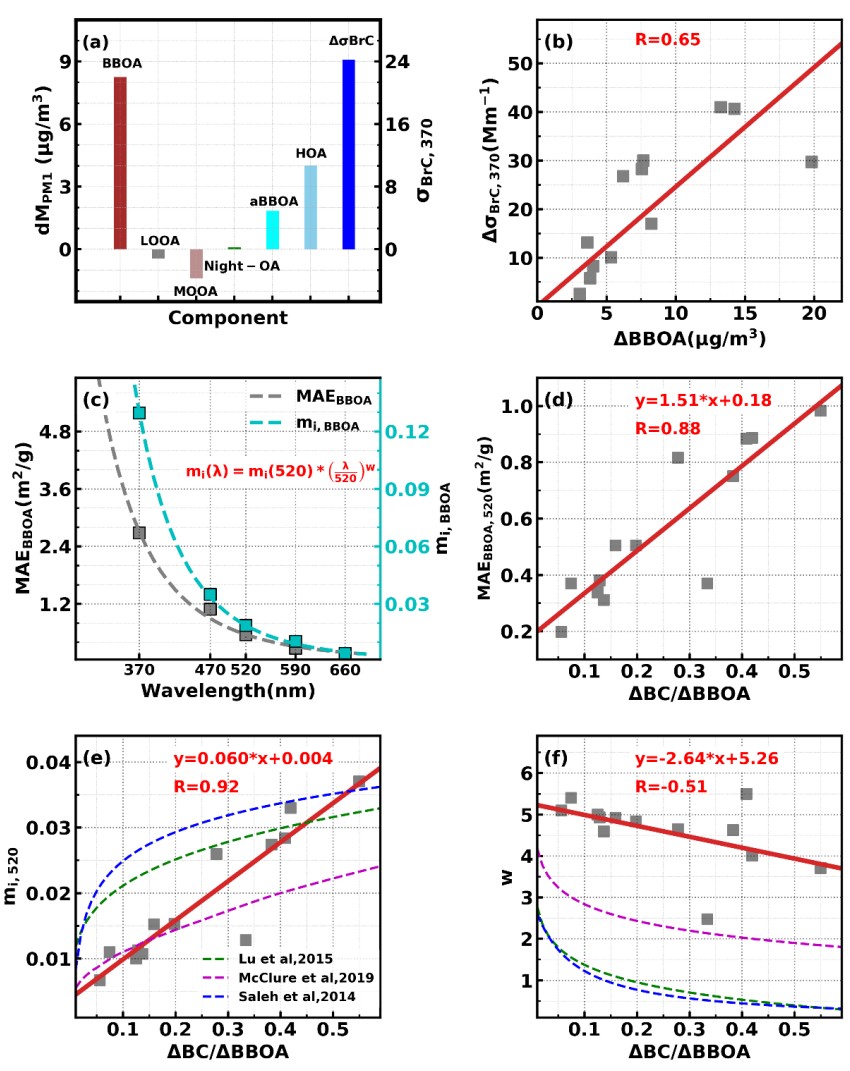

**Figure 5.** **(a)** Average changes of organic aerosol components for BBOA spikes when BC measurements are available; **(b)** Relationships between derived $\Delta\sigma_{BrC}$ at 525 nm only associated with BBOA and $\Delta$ BBOA; **(c)** Average spectral dependence of $MAE_{BBOA}$ and BBOA $m_i$ ; **(d)** Relationship between $MAE_{BBOA}$ at 525 nm and $\Delta CO/\Delta BBOA$; **(e)** Relationship between BBOA $m_i$ at 525 nm and $\Delta CO/\Delta BBOA$; **(f)** Relationship between the spectral dependence parameter w of BBOA $m_i$ and $\Delta CO/\Delta BBOA$.

Derived BrC absorptions of BBOA spikes were used to calculate $MAE_{BBOA}$ and retrieve
imaginary part of BBOA refractive index ($m_{i,BBOA}$) in combination of retrieved BBOA size
distributions using Mie theory. Average changes of organic aerosol components for spikes with



available $\sigma_{BrC}$ values are shown in Fig.5a. ΔBBOA dominated the mass changes, however, non-
negligible changes for aBBOA, HOA and MOOA. The average MAE$_{HOA}$, MAE$_{aBBOA}$ and MAE$_{MOOA}$
are estimated using multilinear regression for all data points with values at 370 nm of 0.1, 0.96 and 0.9
m$^2$/g, respectively. Thus the $\Delta\sigma_{BrC,BBOA}$ can be derived as $\Delta\sigma_{BrC,BBOA}(\lambda)=\Delta\sigma_{BrC,derived}$-ΔHOA×
MAE$_{HOA}(\lambda)$-ΔaBBOA×MAE$_{aBBOA}(\lambda)$–ΔMOOA×MSE$_{MOOA}(\lambda)$. As shown in Fig.5b, $\Delta\sigma_{BrC,BBOA}$
was moderately correalted with ΔBBOA (R=0.65), suggesting significant changes of MAE$_{BBOA}$.
Derived MAE$_{BBOA}$ exhibited strong wavelength dependence and average values at wavelengths of 370,
470, 520, 590, and 660 nm were 2.46,0.99,0.53,0.28, 0.11 m$^2$/g, respectively. Fig.5c shows the spectral
dependence of MAE$_{BBOA}$ and retrieved m$_{i,BBOA}$, and formula form that parameterize the spectral
dependence was consistent with previous studies (Saleh et al., 2014). BBOA absorption properties
depended largely on combustion conditions, both MAE$_{BBOA}$ and retrieved m$_i$ at 520 nm was highly
and linearly correlated with ΔBC/ΔBBOA (Fig.5d and Fig.5e). Results regarding m$_{i,BBOA}$
parameterizations as a function of ΔBC/ΔBBOA of previous studies are also shown in Fig.5e. Results
of Saleh et al. (2014) and Lu et al. (2015) at 550 nm were higher for ΔBC/ΔBBOA in the range of 0.05
to 0.4. Curve of McClure et al. (2020) well described the m$_{i,BBOA}$ variations for ΔBC/ΔBBOA less than
0.2. The m$_{i,BBOA}$ spectral dependence parameter w$_{BBOA}$ ranged from 2.5 to 5.5 with an average of 4.7,
was linearly and negatively correlated to ΔBC/ΔBBOA and much higher than those reported in Saleh
et al. (2014) and Lu et al. (2015). The w$_{BBOA}$ was also higher than the fitted line of McClure et al.
(2020), however, was actutally consistent with the w$_{BBOA}$ range reported in Fig.5c of McClure et al.
(2020) for a BC/OA range of 0.1 to 0.55.

**4. Implications for simulating climate effects of BBOA**
Findings of BBOA size distributions, real and imginary parts of BBOA refractive index in this
study have important implications for climate modelling of BBOA radiative effects. The volume
dominant mode of biomass burning aerosols contribute dominantly to aerosol mass, which are most
important for BBOA scattering and absoprtion properties. The volume dominant mode also contributed
dominantly to number concentration for diameter range of >150 nm, and this diameter range played
the dominant role in BBOA aerosols as cloud condensation nucei (Chen et al., 2019). However,
previous studies usually parameterized number geometric mean diameter D$_{gn}$ as a function of



combustion conditions. It was found that BBOA mass scattering efficiency correlated well with the
volume geometric mean diameter $D_{gv}$, but correlated poorly with $D_{gn}$, which was in contradiction with
previous results (Levin et al., 2010;Laing et al., 2016) that BBOA mass scattering efficiency was highly
correlated with $D_{gn}$. However, the simulation results shown in Fig.S10 explained the contrast, that
aerosol scattering efficiency were very sensitive to $\sigma_g$ changes for fixed $D_{gn}$, however, are much less
sensitive to $\sigma_g$ changes for $D_{gv}$, and retrieved $\sigma_g$ varied over a wide range from 1.2 to 2 in this study.
In addition, it was found that $D_{gn}$ correlated poorly with normalized parameters such as $\Delta CO/\Delta BBOA$,
whereas $D_{gv}$ correlated highly with $\Delta CO/\Delta BBOA$ . Therefore, representing BBOA volume size
distribution of the volume dominant mode as a function of combustion conditions in climate models
might be a better choice if using only one size distribution mode (Stier et al., 2005;Dentener et al.,
2006), however needs further and synthesized research on this topic. In view of this, on the basis of
the relationships between $\Delta CO/\Delta BBOA$ and $\Delta BC/\Delta BBOA$, the $D_{gv}$ were parameterized as $D_{gv}=632-$
$1000\times\Delta BC/\Delta BBOA$, and might be applicable in climate models (Saleh, 2020b).

The real part of BBOA refractive $m_{R,BBOA}$ was fundamental parameter for simulating BBOA

scattering properties in Climate models, however, a constant was usually used due to the lack of
adequate parameterizations (Brown et al., 2021). Significant changes were found in $m_{R,BBOA}$ in this
study (1.47 to 1.64), and the variations were likely closely associated with changes in fire combustion
conditions represented by $\Delta CO/\Delta BBOA$. For BBOA refractive index, the imaginary part ($m_{i,BBOA}$) are
currently recommended to be parameterized as a function of BC/BBOA ratio (Saleh et al., 2014), which
is supported by results of several studies (Lu et al., 2015;McClure et al., 2020). Results of this study
suggests that it might be also feasible to parameterize $m_{i,BBOA}$ as a function of BC/BBOA, however,
needs further comprehensive investigations.

The immaginary part of BBOA refractive index, $m_{i,BBOA}$, plays crucial role in representing BBOA

absorptivity in climate models. Linear relationships between $m_{i,BBOA}$ as well as the spectral dependence
parameter w and BC/OA are reported for the first time in this study. The observed BC/OA ratio (0.05
to 0.55) locates within the upper range of previously reported BC/OA values. Few measurements
regarding aerosol refractive index and size-distributions are available in this BC/OA range, and no
researches have focused on parameterizations of BBOA refractive index in this specific BC/OA range,
thus results of this study have partially filled this gap. Results of McClure et al. (2020) demonstate
that a sigmoidal curve fitts well the $m_{i,BBOA}$ variations for a wide range of BC/OA ratio ($10^{-5}$ to 10),



however the $m_{i,BBOA}$ variations are not well captured by the fitted curve for BC/OA>0.1. We
recommend for more sophisticated parameterizations of $m_{i,BBOA}$ under different BC/OA ranges.


**Data availability**. The data used in this study are available from the corresponding author upon request
Ye Kuang (kuangye@jnu.edu.cn) and Shan Huang (shanhuang_eci@jnu.edu.cn)
**Competing interests**. The authors declare that they have no conflict of interest.

**Author Contributions**.
YK and SH designed this experiment, YK conceived and led this research. BL and YK wrote the
manuscript. SH lead the SP-AMS measurements and particle number size distribution measurements.
SH performed the PMF analysis and Cx fragment analysis, revised the manuscript. MS and BY planned
this campaign. DC and DY provided authority of conducting the campaign in Heshan supersite and
gave data availability from the site. All other coauthors have contributed to this paper in different ways.
**Acknowledgments**
This work is supported by the National Natural Science Foundation of China (grant No. 41805109,
41807302), National Key Research and Development Program of China (grant No. 2017YFC0212803,
2016YFC0202206), Key-Area Research and Development Program of Guangdong Province (grant No.
2019B110206001), Special Fund Project for Science and Technology Innovation Strategy of
Guangdong Province (grant No.2019B121205004), Guangdong Natural Science Funds for
Distinguished Young Scholar (grant No. 2018B030306037) and Guangdong Innovative and
Entrepreneurial Research Team Program (grant No. 2016ZT06N263).

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
