# Peer review of "Parametrizations of size distribution and refractive index of biomass burning organic 1 aerosol with black carbon content 2 Biao Luo1,2, Ye Kuang1,2,\*, Shan Huang1,2\*, Qicong Song1,2, Weiwei Hu3, Wei Li1,2, Yuwen Peng1,2,"

_Atmospheric Chemistry and Physics, 2022_

## Referee Comment (RC1)

This manuscript shows an improved method for estimating the optical properties of BrC and their size distribution, which are one of the biggest challenges in estimating their climate effects. This study will be an interesting contribution to our understanding of BrC's climate effects. However, I still have some comments to help the author improve this manuscript. In general, I recommend that the editor consider this manuscript as a major revision.

**Major comments:**

1. Acronyms and abbreviations need to be well defined the first time they appear in the manuscript. Please check that.
2. I am not a big fan of calculating AAE by only using two wavelengths. It can increase the influence of systematic errors due to measurement. Moreover, usually, AAE calculated using short-wavelength will be larger since they are more dominating than longer wavelengths. In that case, I suggest using power law fitting since you have more absorption measurements at 7 wavelengths. Please consider that in future studies and add some discussion in the manuscript.
3. I did not see a particle scattering correction for aethalometer measurement, although you have particle scattering measurement. The correct factor you used might not work well for your sample. I want to see some discussion of this issue.
4. Correct me if I am wrong. What wavelengths do you use to calculate BrC absorption? I might miss that in your manuscript.
5. I did not see any comparison between modeled results and standards such as direct absorption measurements of BC (or any BC surrogate such as cab-o-jet, see "Characterization of light-absorbing aerosols from a laboratory combustion source with two different photoacoustic techniques") and coated. Without these experiments, it is not easy to validate your method. Have you considered performing these types of experiments?
6. BrC can also absorb light near-IR (see "Characterization of light-absorbing aerosols from a laboratory combustion source with two different photoacoustic techniques"; "Investigating the dependence of light absorption properties of combustion carbonaceous aerosols on combustion conditions"). Thus, I suggest you should also add some discussion about BrC absorption at near-IR

**Specific comments**
7. L141-143, "including multi-wavelength … 2021b)." This is not clear to me. Could you provide more details about your scattering measurements? I want to know what type of instrument you used and the wavelength of that instrument.
8. L171-173, "The mass concentrations … components." This part is also not clear to me. How do you do offline filter measurements and online inorganic aerosol component measurements? If you have any data, I suggest including them in the SI to support your argument.
9. L175-186, "Six-factors … (2021b)." This part is a little bit confusing to me. Are these thresholds developed before, or are these just the average value of each class? Also, in Fig. S3, you showed more element ratios such as N:C, H:X, and OM:OC. I am also not clear on how did you get these values. I suggest putting these parameters in a table.
10. L186-188, "On the basis … this study." With your current setup, it should be able to retrieve the density of BBOA and HOA by using SP-AMS. I am curious how close these values are to the literature values.
11. Saleh, 2020a and Saleh, 2020b are the same. Please correct that.
12. L231-235, "These parameters … mass size distributions." This part is not clear to me. Did you use ranges of these parameters to calculate the AAE? Then what are the ranges you used? How did you decide on the ranges?

13. L254-257, "The average … respectively." First, the first one and the last one are the same. Please check and correct that. Second, what is the uncertainty range of these ratios? Are they more significant than the uncertainty? These ratios are very close and increase with increasing of λ. This might be due to the increased weight of absorption at short wavelengths.

14. L296-298. "During the … occurred." It is not clear to me how you chose spikes. In Fig. S7, the Shaded areas are very difficult to see. Please consider using a different color. Some BBOA spikes are not highlighted (e.g., beginning of Oct 19 and end of Oct 23). Is there any reason for that?

15. L314-316, "The average … 2020)." How do you calculate Δ for all parameters you show in this manuscript? This is not clear to me. I assume the Δ you used in the manuscript is the difference between that variable before and after the BBOA spike. Then, my question is, what are the start and end times you used to get the average before and during the spike? This is not clear to me and can significantly affect your results.

16. L329-333, "The $D_{gv}$ … 2000)." Please check these two sentences carefully. In L329, you subscripted $D_a$. In L 330, you used $D_{va}$ instead of $D_a$, which $D_{va}$ should be more suitable. In L332, you used C as the factor. You need to use a different letter since you previously defined C as the Multiple-scattering correction factor.

17. L345-347, "BC/BBOA ratio … R=0.84)." Is CO in L346 carbon monoxide? If yes, do you also have CO2 measurements? Then you can use modified combustion efficiency (MCE) to estimate combustion efficiency. Also, please describe CO and CO2 measurements in your Method section.

18. L466-468, "The average … respectively." This is not clear to me. How did you do that? How do you get absorption for HOA, aBBOA, and MOOA?

19. L470, "suggesting significant changes of $MAE_{BBOA}$." This is not clear to me. What are the significant changes you mentioned here, and why are there significant changes? Please explain that to me.

20. L474-475, The results in this section can also be supported by "Investigating the dependence of light-absorption properties of combustion carbonaceous aerosols on combustion conditions ", "Brownness of organics in aerosols from biomass burning linked to their black carbon content", "Light-Absorbing organic carbon from prescribed and laboratory biomass burning and gasoline vehicle emissions", and "Parameterization of single-scattering albedo (SSA) and absorption Ångström exponent (AAE) with EC / OC for aerosol emissions from biomass burning". Please consider adding these references.

21. In SI, L73, How do you calculate the mass fraction of pure externally mixed BC?

22. In SI, L74, R_NBC is not well defined. Please add details like how you retrieve it.

23. In SI, L87, the density of BC should use 1.8 g cm$^{-3}$ from "Bounding the role of black carbon in the climate system: A scientific assessment", unless you measured the BC density or have other references.

24. In SI, L116, what is Fig. Sx? I did not see it.

---

## Author Response (AR1)

**Dear Editor:**

We are grateful for the reviewer's careful inspection of our manuscript. We have addressed all concerns raised by Referee #1, and re-organized the abstract as suggested by Referee #2 with the current version focus more on the most fundamental findings. The supplement and manuscript are also re-organized to make it easier to read, and as suggested by Referee #2, information on choosing the best number of PMF factors, on diagnostics of the statistical model, on the interpretation of the factors were also added in the supplement. We really appreciate valuable comments and suggestions raised by the two reviewers which improved our manuscript. All these comments raised by the referees have been explicitly replied point by point and incorporated into the revision. We believe that the revised manuscript is now more convincing than before.

**Responses to anonymous referee #1**

**Major comments:**

**Comment:** Acronyms and abbreviations need to be well defined the first time they appear in the manuscript. Please check that.

**Response:** Many thanks. We scrutinized the manuscript and made sure that all Acronyms and abbreviations have defined adequately when they appear first in the manuscript, and repeated definitions such as BBOA, AAE and PMF are revised, and the definitions of NOx, CO, etc are added.

**Comment:** I am not a big fan of calculating AAE by only using two wavelengths. It can increase the influence of systematic errors due to measurement. Moreover, usually, AAE calculated using short-wavelength will be larger since they are more dominating than longer wavelengths. In that case, I suggest using power law fitting since you have more absorption measurements at 7 wavelengths. Please consider that in future studies and add some discussion in the manuscript.

**Response:** Many thanks, we understand that calculating AAE use two wavelengths bear some uncertainties due to measurement errors. However, we cannot agree with the reviewer that measurements at 7 wavelengths should be used: (1) as discussed in Sect 2.3 of the manuscript and pointed out by results of several previous studies, even for pure BC, the AAE is wavelength dependent and the spectral dependence of BC must be took into account when deriving BrC absorptions at multi-wavelengths ; (2) For periods with substantial contribution of BrC absorptions to total aerosol absorptions, the AAE of BC absorptions fitted with absorption measurements of 7 wavelengths will be significantly biased due influences of BrC absorptions .

As to the influences of measurements errors on derivation of $AAE_{BC}$. The following discussions is added the Sect.2.3:

"In addition, the key part of our newly proposed method is considering the spectral dependence of $AAE_{BC}$ through the ratio $R_{AAE}$ and $AAE_{BC,950-880}$, however, the accurate $AAE_{BC,950-880}$ derivations need robust performance of AE33 at both 880 nm

and 950 nm, quality assurance of these measurements should be warranted before using the $AAE_{BC,950-880}$."

**Comment:** I did not see a particle scattering correction for aethalometer measurement, although you have particle scattering measurement. The correct factor you used might not work well for your sample. I want to see some discussion of this issue.

**Response:** Many thanks, we agree with the reviewer that more discussions should be included. The scattering correction for aethalometer measurements were done through $b_{ATN}/C$ with C as the scattering correction factor. Results of previous studies demonstrate C are influenced both by filter type and aerosol chemical composition (Drinovec et al., 2015;Collaud Coen et al., 2010;Yus-Díez et al., 2021). Results of Yus-Díez et al. (2021) showed that C values increased considerably when SSA is higher than 0.95, however, much lower than 0.95 during this field campaign as shown in Fig.1 and the average SSA is 0.78. Thus, discussions about the C value is revised as:

"C is considered to be dependent on filter tape (Drinovec et al., 2015) and aerosol chemical compositions (Wu et al., 2009;Collaud Coen et al., 2010). Results of Yus-Díez et al. (2021) showed that C values increased considerably when SSA is higher than 0.95. However, as shown in Fig.S5, SSA is much lower than 0.95 during this field campaign with an average of 0.78. Moreover, the filter tape 8060 was used for AE33 during this field campaign. Zhao et al. (2020) evaluated C of filter tape 8060 through comparing AE33 measurements with a three-wavelength photoacoustic soot spectrometer, and their results demonstrated that C is almost independent of wavelength and differs little among measurements of different locations. Thus the wavelength independent C of filter tape 8060 of 2.9 recommended by Zhao et al. (2020) was used, and this value is also almost the median value of C ranges used in Kasthuriarachchi et al. (2020)."

[Figure]

**Figure 1**. Time series of SSA during the campaign.

**Comment:** Correct me if I am wrong. What wavelengths do you use to calculate BrC absorption? I might miss that in your manuscript.

**Response:** Many thanks for pointing this out, this comment urged us to make it clearer about the derivation method of BrC absorptions and used wavelengths.

The philosophy of our newly proposed method is introduced briefly here. Results of previous studies (Wang et al., 2018;Li et al., 2019a) demonstrated that significant wavelength dependence of $AAE_{BC}$ and constant assumption of $AAE_{BC}$ in BrC absorption retrievals would bring some uncertainties. In this study, we introduce a AAE ratio $R_{AAE}=AAE_{BC,\lambda-880}/AAE_{BC,950-880}$ to take spectral dependence of $AAE_{BC}$ into account and use on-line measurements of $AAE_{950-880}$ as $AAE_{BC,950-880}$ under the assumption of that negligible absorption contributions of BrC at wavelengths of 880 nm and 950 nm. Thus, absorption measurements of 370 nm, 470 nm, 530 nm, 590 nm and 660 nm can be used to retrieved the spectral dependence of BrC absorptions.

To make this clearer in the manuscript, we have added the following sentences in the method part:

" In this study, we introduce a AAE ratio $R_{AAE}(\lambda)=AAE_{BC,\lambda-880}/AAE_{BC,950-880}$ to take spectral dependence of $AAE_{BC}$ into account and use on-line measurements of $AAE_{950-880}$ as $AAE_{BC,950-880}$ under the assumption of that negligible absorption contributions of BrC at wavelengths of 880 nm and 950 nm. Thus, absorption

measurements of 370 nm, 470 nm, 530 nm, 590 nm and 660 nm can be used to retrieved the spectral dependence of BrC absorptions.

$R_{AAE}(\lambda)$ are influenced by many factors such as BC refractive index, coating shell refractive index as well as BC mixing state, and BC mass size distributions (Li et al., 2019b). A sensitivity experiment following the method of Li et al. (2019b) is initiated to explore impacts of these optical and mixing state parameters on $AAE_{BC,\lambda-880}$ and $R_{AAE}(\lambda)$, more details are available in Supplement Sect. S1…"

**Comment:** I did not see any comparison between modeled results and standards such as direct absorption measurements of BC (or any BC surrogate such as cab-o-jet, see "Characterization of light-absorbing aerosols from a laboratory combustion source with two different photoacoustic techniques") and coated. Without these experiments, it is not easy to validate your method. Have you considered performing these types of experiments?

**Response:** We agree with the reviewer that the validation of $AE_{33}$ measurements is very important.

For AE33 measurements, the loading effect was automatically corrected using dual-spot mode measurements, theoretically, the absorption measurements by AE33 are associated mostly with correction of multiple scattering effect. As discussed in the manuscript and responses before, SSA during observation period is much lower than 0.95 thus the filter type plays the dominant role in determining the scattering correction factor C. Previous results of Zhao et al. (2020) evaluated C of the filter tape we used through comparing AE33 measurements with a three-wavelength photoacoustic soot spectrometer, and their results demonstrated that C is almost independent of wavelength for the filter tape we used (filter tape 8060) and differs little among measurements of

different locations. Thus, the recommended C value of 2.9 by Zhao et al. (2020) is used

[Figure]

**Figure 2**. Comparisons between aerosol absorptions at 880 nm measured by AE33 and measurements and simulated BC absorptions at 880 nm using measurements of DMA-SP2 system.

in this study. Even so, we still worry about the data quality of the AE33 measurements by using a C value of 2.9. To test the robustness of the recommended C of filter tape 8060 by Zhao et al. (2020), we conducted field measurements in winter of 2022 in Guangzhou urban area (the distance between this new site and the site of this study is about 100 km) using both AE33 (same filter type and C with this study) and an DMA-SP2 system which measures the mass size distribution and mixing state of BC for diameter range of 80 nm to 700 nm. As shown in Fig.2, an excellent agreement between aerosol absorptions at 880 nm measured by AE33 and simulated BC absorptions at 880 nm using measurements of DMA-SP2 system, which boosted our confidence about aerosol absorption measurements in this study. Of course, direct measurements of aerosol absorption measurements using photoacoustic techniques would be more convincing, thus we considered perform experiments recommended by the reviewer in future field campaigns.

**Comment:** BrC can also absorb light near-IR (see "Characterization of light-absorbing aerosols from a laboratory combustion source with two different photoacoustic

techniques"; "Investigating the dependence of light absorption properties of combustion carbonaceous aerosols on combustion conditions"). Thus, I suggest you should also add some discussion about BrC absorption at near-IR.

**Response:** Many thanks, we agree with the reviewer that discussions about BrC absorptions at near-IR spectral ranges should be added. The following sentences are added in Sect2.3 of the manuscript:

"Results of previous studies (Saleh, 2020;Yu et al., 2021) demonstrated that non-negligible BrC absorptions at near-infrared range, and results of Hoffer et al. (2017) demonstrated that absorption coefficient of tar balls at 880 nm is more than 10% of that at 470 nm. During this campaign, the average aerosol absorption at 880 nm is 26.7 Mm-1, derived average BrC absorption at 470 nm is 11.4 Mm-1, 10% of BrC absorption at 470 nm accounts for on average 4.2% of aerosol absorption at 880 nm and the realistic BrC contribution at 880 nm is likely lower considering that tar balls represent the most efficient BrC. Thus, the assumption that negligible absorption contributions of BrC at wavelengths of 880 nm and 950 nm when deriving $AAE_{BC,950-880}$ from AE33 measurements holds in most cases when BC dominates."

**Technical points:**

**Comment:** L141-143, "including multi-wavelength … 2021b)." This is not clear to me. Could you provide more details about your scattering measurements? I want to know what type of instrument you used and the wavelength of that instrument.

**Response:** Instrument type and wavelengths are added. As the following:

"including multi-wavelength (450 nm, 525 nm, 635 nm) aerosol scattering coefficients (nephelometer, Aurora 3000 )…"

**Comment:** L171-173, "The mass concentrations … " This part is also not clear to me. How do you do offline filter measurements and online inorganic aerosol component measurements? If you have any data, I suggest including them in the SI to support your argument.

**Response:** These aspects are carefully discussed in supplement of a previous paper (Kuang et al., 2021), including them here in the supplement would made the supplement redundant. To make this clearer, this part is revised as: "As discussed in Kuang et al. (2021), the mass concentrations of aerosol chemical compositions from SP-AMS were validated by offline PM2.5 filter measurements, SMPS aerosol volume concentration measurements and online measurements for inorganic aerosol components."

**Comment:** "Six-factors … (2021b)." This part is a little bit confusing to me. Are these thresholds developed before, or are these just the average value of each class? Also, in Fig. S3, you showed more element ratios such as N:C, H:X, and OM:OC. I am also not clear on how did you get these values. I suggest putting these parameters in a table. L186-188, "On the basis … this study." With your current setup, it should be able to retrieve the density of BBOA and HOA by using SP-AMS. I am curious how close these values are to the literature values.

**Response:** We do not understand what "these thresholds" refer to? The source apportionment of organic aerosols using PMF on the basis of AMS measurements is a common and well-developed technique in the aerosol chemistry community. Considering that all the other elaborations made in the present manuscript are based on the determination of the BBOA factor. In the revised manuscript, the section "determination of PMF factors from SP-AMS measurements" was added in the supplement as Sect S1.1. In this section, information on choosing the best number of factors, on diagnostics of the statistical model, on the interpretation of the factors were added, as well as profiles and time-series of those factors. Elemental ratios such as N:C, H:X, and OM:OC are not used in the manuscript, thus we thought these parameters in a table is not necessary and put them together with OA factor spectral is more convenient for readers who cares details of OA factors.

Actually, we don't understand how densities of BBOA and HOA can de retrieved using SP-AMS measurements, we thought densities of organic aerosols is beyond the measurement capacity of SP-AMS measurements. SP-AMS provides only elemental ratios of OA factors through PMF technique, with these elemental ratios, densities of

OA factors can be calculated on the basis of previous OA density parameterization schemes. This is what we have done, the scheme proposed by Kuwata et al. (2012), based on our knowledge, is the only available one regarding OA density calculation on the basis of elemental ratios (please correct us if the reviewer have any other clues). Also, we do not find any available direct measurements of BBOA and HOA densities, thus comparing existing literature values is not practicable (please let us know if the reviewer has any).

**Comment:** Saleh, 2020a and Saleh, 2020b are the same. Please correct that.

**Response:** Thanks, corrected.

**Comment:** L231-235, "These parameters … mass size distributions." This part is not clear to me. Did you use ranges of these parameters to calculate the AAE? Then what are the ranges you used? How did you decide on the ranges?

**Response:** Yes we used ranges of these parameters to calculate $AAE_{BC,370-880}$ and $R_{AAE}(370)$ ($AAE_{BC,370-880}/AAE_{BC,950-880}$). To be clearer, following sentences are added:

"Impacts of these parameters on $AAE_{BC,370-880}$ and $R_{AAE}(370)$ are investigated through perturb the parameter within atmospheric relevant ranges reported in previous studies (Bond et al., 2013;Tan et al., 2016;Zhao et al., 2019), and ranges of these parameters are listed in Fig.1"

**Comment:** L254-257, "The average … respectively." First, the first one and the last one are the same. Please check and correct that. Second, what is the uncertainty range of these ratios? Are they more significant than the uncertainty? These ratios are very close and increase with increasing of λ. This might be due to the increased weight of absorption at short wavelengths.

**Response:** Thanks, the second should be $AAE_{BC,\,470-880}/AAE_{BC,\,950-880}$, and the last one should be deleted. The average and standard deviations of these ratios are 0.79(±0.044), 0.85(±0.038), 0.88(±0.035), 0.9(±0.035) and 0.93(±0.031). These ratios

seem very close, but are significant, for example, if $AAE_{BC, 950-880}$ is 1.2, then 0.79 of the ratio $AAE_{BC, 370-880}/AAE_{BC, 950-880}$ corresponds to $AAE_{BC, 370-880}$ of 0.95. Standard deviations of these parameters are added in the manuscript.

**Comment:** L296-298. "During the … occurred." It is not clear to me how you chose spikes. In Fig.S7, the Shaded areas are very difficult to see. Please consider using a different color. Some BBOA spikes are not highlighted (e.g., beginning of Oct 19 and end of Oct 23). Is there any reason for that?

**Response:** Events with BBOA increased suddenly, drastically and continuously within half hour to several hours were identified as BBOA spikes. Thus, we don't have a criterion on this and we choose spikes artificially, the actual last times of identified spikes generally range from 1-3 hours. The used BBOA spikes were shown in Fig.S8, some of identified spikes were used because of the missing of particle number size distribution measurements. The shading color was changed and peak BBOA points were also marked in Fig.S8. To make this point clearer, following sentences are added in Sect 3.2.:

"During the observation period, BBOA contributed domominantly to BrC absorptions and notable biomass burning events represented by BBOA mass concentration spikes as shown in Fig.S8 freqeuntly occurred. Events with BBOA increased suddenly, drastically and continuously within half hour to several hours were identified as BBOA spikes. we don't have a criterion on this and we choose spikes artificially, these identified spikes generally last about 1-3 hours (from the beginning to the peak). The used BBOA spikes were shaded in Fig.S8, some of identified spikes were not used because of the missing of particle number size distribution measurements."

**Comment:** L314-316, "The average … 2020)." How do you calculate Δ for all parameters you show in this manuscript? This is not clear to me. I assume the Δ you used in the manuscript is the difference between that variable before and after the BBOA spike. Then, my question is, what are the start and end times you used to get the average before and during the spike? This is not clear to me and can significantly affect

your results.

**Response:** The Δ we used corresponds to the difference between that variable before BBOA increases and when BBOA reach its peak (the definition of the BBOA spike), corresponding to the start and end of BBOA increase. To make this clearer, the following sentence is added:

"Note that the Δ shown in Fig.3a and also hereafter means the difference between that variable before BBOA increases and when BBOA reach its peak (the definition of the BBOA spike, these peaks are also marked in Fig.S8), corresponding to the start and end of BBOA increase."

**Comment:** L329-333, "The Dgv … 2000)." Please check these two sentences carefully. In L329, you subscripted Da. In L 330, you used Dva instead of Da, which Dva should be more suitable. In L332, you used C as the factor. You need to use a different letter since you previously defined C as the Multiple-scattering correction factor.

**Response:** Thanks, corrected and a new Cs is defined.

**Comment:** L345-347, "BC/BBOA ratio … R=0.84)." Is CO in L346 carbon monoxide? If yes, do you also have CO2 measurements? Then you can use modified combustion efficiency (MCE) to estimate combustion efficiency. Also, please describe CO and CO2 measurements in your Method section.

**Response:** CO is carbon monoxide, but unfortunately, we don't have CO2 measurements. The CO measurements is added in the Method section.

**Comment:** L466-468, "The average … respectively." This is not clear to me. How did you do that? How do you get absorption for HOA, aBBOA, and MOOA?

**Response:** The multilinear regression between derived BrC absorptions and OA factors were usually used in previous studies to obtain average MAC values of OA factors such

as de Sá et al. (2019), and also used in this study as shown in Fig.3.

[Figure]

Figure 3. Multilinear fitting of BrC absorptions at 370 nm with OA factors

To make this clearer, this sentence is revised as :

"The average MAE$_{HOA}$, MAE$_{aBBOA}$ and MAE$_{MOOA}$ are estimated using multilinear regression for all data points as shown in Fig.S11 with values at 370 nm of 0.1, 0.96 and 0.9 m$^2$/g, respectively. "

**Comment:** L470, "suggesting significant changes of MAEBBOA." This is not clear to me. What are the significant changes you mentioned here, and why are there significant changes? Please explain that to me.

**Response:** "significant changes" is not a suitable description. As shown in Fig.5b, $\Delta\sigma_{BrC,BBOA}$ correlates moderately with $\Delta$BBOA (R=0.65) which means that the ratio MAE$_{BBOA}$=$\Delta\sigma_{BrC,BBOA}$ /$\Delta$BBOA differs much among identified biomass burning plumes. To make this clearer, this sentence is revised as:

"As shown in Fig.5b, $\Delta\sigma_{BrC,BBOA}$ was moderately correalted with $\Delta$BBOA (R=0.65), suggesting MAE$_{BBOA}$=$\Delta\sigma_{BrC,BBOA}$/$\Delta$BBOA differs much among identified plumes."

**Comment:** L474-475, The results in this section can also be supported by "Investigating the dependence of light-absorption properties of combustion carbonaceous aerosols on combustion conditions ", "Brownness of organics in aerosols from biomass burning linked to their black carbon content", "Light-Absorbing organic carbon from prescribed and laboratory biomass burning and gasoline vehicle emissions", and "Parameterization of single-scattering albedo (SSA) and absorption Ångström exponent (AAE) with EC/OC for aerosol emissions from biomass burning". Please consider adding these references.

**Response:** Thanks, references have been added. This sentence is revised as:

"BBOA absorption properties depended largely on combustion conditions, consistent with results of previous studies (Saleh et al., 2014;Lu et al., 2015;Pokhrel et al., 2016;Xie et al., 2017;Cheng et al., 2019;McClure et al., 2020), both $MAE_{BBOA}$ and retrieved $m_i$ at 520 nm was highly and linearly correlated with $\Delta BC/\Delta BBOA$ (Fig.5d and Fig.5e)."

**Comment:** In SI, L73, How do you calculate the mass fraction of pure externally mixed BC?

**Response:** We didn't calculate it, we perturbed this parameter within the range of 0.2 to 0.8 in the sensitivity experiments introduced in Sect 2.3. When retrieving BrC absorptions, calculations of $R_{AAE}(\lambda)$ were achieved through assuming an $r_{ext}$ of 0.5 due to the small influences of $r_{ext}$ variations on $R_{AAE}(\lambda)$. The following sentence is added in the SI:

"When retrieving BrC absorptions, calculations of $R_{AAE}(\lambda)$ were achieved through assuming $r_{ext}$ and R_NBC values of 0.5 and 0.5 due to small influences of $r_{ext}$ and R_NBC variations on $R_{AAE}(\lambda)$."

**Comment:** In SI, L74, R_NBC is not well defined. Please add details like how you retrieve it.

**Response:** Thanks, the definition of R_NBC has been added in the supplement. We didn't retrieve R_NBC, we perturbed this parameter within the range of 0.1 to 0.7 in

the sensitivity experiments introduced in Sect 2.3. When retrieving BrC absorptions, calculations of $R_{AAE}(\lambda)$ were achieved through assuming an R_NBC of 0.5 due to the small influences of $r_{ext}$ variations on $R_{AAE}(\lambda)$. The following sentence is added in the SI:

"When retrieving BrC absorptions, calculations of $R_{AAE}(\lambda)$ were achieved through assuming $r_{ext}$ and R_NBC values of 0.5 and 0.5 due to small influences of $r_{ext}$ and R_NBC variations on $R_{AAE}(\lambda)$."

**Comment:** In SI, L87, the density of BC should use 1.8 g cm-3 from "Bounding the role of black carbon in the climate system: A scientific assessment", unless you measured the BC density or have other references.

**Response:** Thanks, we have added this reference.

**Comment:** In SI, L116, what is Fig. Sx? I did not see it.

**Response:** Thanks, should be Fig.S2a and corrected.

[revised manuscript text omitted]

**Responses to anonymous referee #2**

**General comments:**

**Comment:** Abstract is too detailed and technical. I strongly recommend to re-organize the abstract, summarizing the most fundamental findings and leaving details for main text and conclusions.

**Response:** Many thanks, we agree with the reviewer, and the abstract is revised as the following:

"Biomass burning organic aerosol (BBOA) impacts significantly on climate directly through scattering and absorbing solar radiation and indirectly through acting as cloud condensation nuclei. However, fundamental parameters in the simulation of BBOA radiative effects and cloud activities such as size distribution and refractive index remain poorly parameterized in models. In this study, biomass burning events with high combustion efficiency characterized by high black carbon (BC) to BBOA ratio (0.22 on average) were frequently observed during autumn in the Pearl River Delta region, China. An improved absorption Ångström exponent (AAE) ratio method considering both variations and spectral dependence of black carbon AAE was proposed to differentiate brown carbon (BrC) absorptions from total aerosol absorptions. BBOA size distributions, mass scattering and absorption efficiency were retrieved based on the changes in aerosol number size distribution, scattering coefficients and derived BrC absorptions that occurred with BBOA spikes. Geometric mean diameter of BBOA volume size distribution $D_{gv}$ depended largely on combustion conditions, ranging from 245 to 505 nm, and a linear relationship between $D_{gv}$ and $\Delta BC/\Delta BBOA$ was achieved. Retrieved real part of BBOA refractive index ranges from 1.47 to 1.64, with evidences showing that its variations might depend largely on combustion efficiency, which is rarely investigated in existing literatures however requires further comprehensive investigations. Retrieved imaginary parts of BBOA refractive index ($m_{i,BBOA}$) correlated highly with $\Delta BC/\Delta BBOA$ (R>0.88) but differ much with previous parameterization schemes. The reason behind the inconsistency might be that single formula parameterizations of

$m_{i,BBOA}$ over the whole BC/BBOA range were used in previous studies which might deviate substantially for specific BC/BBOA ranges. Thus, a new scheme that parameterize wavelength-dependent $m_{i,BBOA}$ was presented, which filled the gap for field-based BBOA absorptivity paramterizations of BC/BBOA>0.1. These findings have significant implications for simulating BBOA climate effects and suggest that linking both BBOA refractive index and BBOA volume size dsitrbutions to BC content might be a feasible and a good choice for climate models."

**Comment:** The application of PMF to AMS data should be better described: neither in the main text nor in the supplementary it is described in any way other than by presenting its resulting chosen solution (profiles and time-series of the factors). Not even in the manuscript already published (referred to in P7, L185-186) there is a detailed description of the procedure used to determine the PMF solution presented (no info on choosing the best number of factors, on diagnostics of the statistical model, on the interpretation of the factors, etc.). Considering that all the other elaborations made in the present manuscript are based on the determination of the BBOA factor, I believe that a broader discussion of the PMF approach and of the robustness of the solution is necessary.

**Response:** We agree with the reviewer that these information should be included in the supplement for reader's convenience considering that all the other elaborations made in the present manuscript are based on the determination of the BBOA factor. In the revised manuscript, the section "determination of PMF factors from SP-AMS measurements" was added as Sect S1.1. In this section, information on choosing the best number of factors, on diagnostics of the statistical model, on the interpretation of the factors were added, as well as profiles and time-series of those factors.

**Technical comments:**

**Comment:** P6, L154-155: unclear and perhaps grammatically incorrect sentence, please rephrase.

**Response:** This sentence is revised as:

"However, aerosol absorption values measured by AE33 bear uncertainties associated with loading and multiple scattering effects."

**Comment:** P6, L158: "babs" in the equation should be subscript.

**Response:** corrected.

**Comment:** Consistency between main text and supplementary should be better checked and the Supplementary should be reorganized accordingly. In particular:
-the order of the supplementary sections should follow the main text order: for instance, SP-AMS PMF results (in Sect. S2) should go before the modelling methods (Sect. S1).
-Some Supplementary Figures are not well presented: for instance, in the legend of Fig. S1b is not possible to differentiate the dashed lines and so to understand what the different lines in the graph are representing.
-In the text of Supplementary (at L116) there is a figure referenced as Fig.Sx.
More inconsistencies can be present and should be checked.

**Response:** Many thanks for the suggestion, we have reorganized the supplement and put PMF analysis of SP-AMS measurements in Sect. S1.1 of the manuscript, and made the method part follow the main text order. The legend of the original Fig.S1b is modified and easy to differentiate, the Fig.Sx is also corrected and we have scrutinized the manuscript and the supplement to avoid in consistencies.